# Plan, Verify and Switch: Integrated Reasoning with Diverse X-of-Thoughts

**Tengxiao Liu**[1]*, **Qipeng Guo**[2], **Yuqing Yang**[1], **Xiangkun Hu**[2],
**Yue Zhang**[3]†, **Xipeng Qiu**[1]†, **Zheng Zhang**[2]

[1]School of Computer Science, Fudan University
[2]Amazon AWS AI, [3]School of Engineering, Westlake University
{txliu21, yuqingyang21}@m.fudan.edu.cn, {gqipeng, xiangkhu, zhaz}@amazon.com
xpqiu@fudan.edu.cn, zhangyue@westlake.edu.cn

## Abstract

As large language models (LLMs) have shown effectiveness with different prompting methods, such as Chain of Thought, Program of Thought, we find that these methods have formed a great complementarity to each other on math reasoning tasks. In this work, we propose **XoT**, an integrated problem solving framework by prompting LLMs with diverse reasoning thoughts. For each question, XoT always begins with selecting the most suitable method then executes each method iteratively. Within each iteration, XoT actively checks the validity of the generated answer and incorporates the feedback from external executors, allowing it to dynamically switch among different prompting methods. Through extensive experiments on 10 popular math reasoning datasets, we demonstrate the effectiveness of our proposed approach and thoroughly analyze the strengths of each module. Moreover, empirical results suggest that our framework is orthogonal to recent work that makes improvements on single reasoning methods and can further generalise to logical reasoning domain. By allowing method switching, XoT provides a fresh perspective on the collaborative integration of diverse reasoning thoughts in a unified framework.

## 1 Introduction

The AI community has long sought to achieve automated reasoning (Hewitt, 1969), which is an important component of Artificial General Intelligence (Steunebrink et al., 2016). Mathematical reasoning, as a cognitive skill essential for humans yet challenging for language models, attracts increasing interests and commitment from researchers (Feigenbaum and Feldman, 1963; Wang et al., 2017; Lu et al., 2022).

With the abilities endowed by in-context learning (ICL), Large Language Models (LLMs)

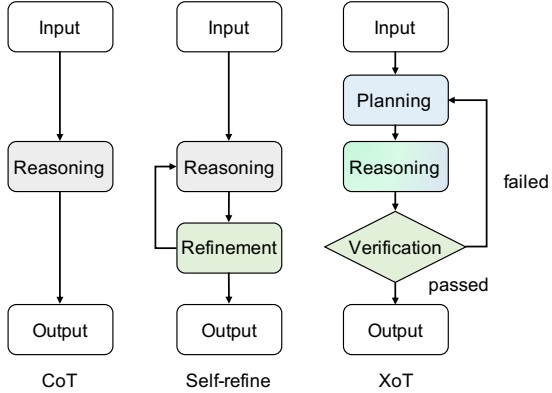

Figure 1: CoT only reasons in a single pass, while self-refine involves refinement using the same method. XoT integrates a verification module that makes a difference in method planning, enabling the attempts of diverse reasoning thoughts within an iterative framework.

(Brown et al., 2020; Chowdhery et al., 2022; Touvron et al., 2023a; OpenAI, 2023) are able to solve mathematical problems through textual rationales with Chain-of-Thought prompting (Wei et al., 2022) (CoT) or through Python functions with Program-Aided Language Model (Gao et al., 2022) and Program-of-Thought prompting (Chen et al., 2022) (PAL or PoT). These prompting methods exhibit unique strengths and limitations. CoT generates a step-by-step reasoning flow in natural language and performs calculations on the fly. This approach enables a more flexible solution format, but may result in a loss of precision since language models often struggle with arithmetic calculations (Lewkowycz et al., 2022; Wei et al., 2022). On the other hand, PoT or PAL resolves problems through Python statements, relying on Python interpreters to ensure calculation accuracy. Another noteworthy and intriguing prompting method is to form math problems as linear equation systems (He-Yueya et al., 2023). Similarly, inspired by Linear Algebra, we propose Equation-of-Thought (EoT), which performs math reasoning in a more direct way.

The diversity inherent in each method does not

---

*Work done during internship at AWS Shanghai AI Lab.
†Corresponding authors.

render them as competing or mutually exclusive alternatives. On the contrary, in practical problem solving scenarios, possessing multiple methods can always yield a range of complementary advantages. The distinct problem-solving approaches can contribute to synergistic benefits that surpass the outcomes of any single approach. We find that this intuition also applies to the realm of math reasoning. With the availability of CoT, PoT and EoT, we hold the hypothesis that a model has the potential to solve a problem if it reaches the correct answer using any one of the prompting methods. As illustrated in Figure 2, our analysis shows that the model exhibits the potential to solve 92.72% of the problems, surpassing the best performing single method by over 10%.

Motivated by this observation, we propose XoT, an integrated math problem solving framework, which improves the LLM's reasoning ability by switching among diverse reasoning thoughts. Since there is no guarantee that LLMs can always solve the problem in a single attempt, we follow the human intuition and allow the model to rethink and switch to a different method when encountering difficulties or obstacles. We apply two complementary verification methods to facilitate the model to decide whether it is time to switch to another method: passive and active verification. Passive verification relies on the external executors to provide determinable results based on the generated programs (Chen et al., 2023; Le et al., 2022). It offers shallow inspections, such as program syntax issues or the runtime errors. For active verification, we ask the model to verify the solution by checking whether the answer adheres to the conditions outlined in the original question.

As shown in Figure 1, XoT consists of three modules that work in an iterative framework: planning, reasoning and verification. Given a problem as input, the planning module first proposes the most appropriate method. The reasoning module then generates one solution using the planned prompting method. With the outputs and the results from external executors, the model is asked to assess the answers in the context of the questions. If the answer fails the verification, we will go back to the planning module for another round of iteration and attempt alternative methods. The iterative process concludes when the verification confirms the correctness of the answer or after exhausting all available methods.

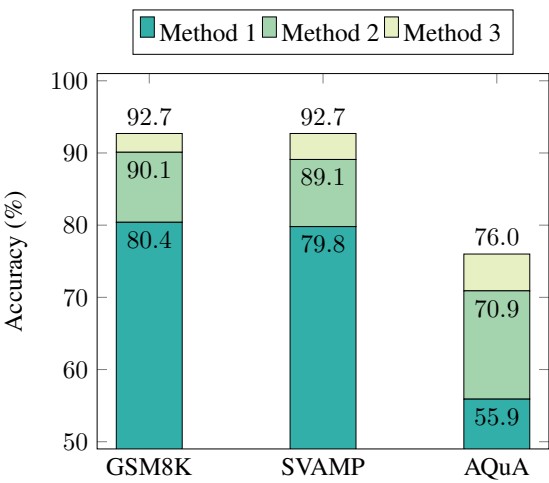

Figure 2: Complementarity of X-of-Thought methods on different datasets. The stacked bars indicate the best performance achieved by using one, two and three methods separately. Employing multiple methods under oracle setting can offer significant performance gains.

To demonstrate the effectiveness of XoT, we conduct extensive experiments on 10 popular mathematical reasoning datasets and achieve consistent improvement. Empirical results suggest that XoT can accommodate recent work that focuses on improving single reasoning methods. Additional experiments also indicate that XoT can generalise to other domains such as logical reasoning tasks.

We summarize the main contributions as follows. First, we propose an integrated problem solving framework XoT, utilising the complementarity of different reasoning thoughts. Second, we introduce EoT which solves math problems with a system of linear equations, serving as a complementary method to existing approaches. Third, we incorporate passive and active verification to facilitate the framework to switch among diverse reasoning thoughts, empowering the framework to make informed decisions regarding the subsequent steps to be taken. More generally, XoT sheds lights on a new direction of interacting with diverse reasoning methods and tools. As shown in Figure 1, instead of sticking to one determined method, LLMs can benefit from the verification and the flexible switching among available reasoning thoughts. [1]

## 2 Related Work

### 2.1 Math Reasoning with LLMs

As the field of large language models continues to prosper, many prompting techniques have emerged

---

[1] Code is publicly available at: https://github.com/tengxiaoliu/XoT.

to unlock the reasoning abilities of LLMs (Qiao et al., 2022). Early success includes reasoning with step-by-step chain of thought (Wei et al., 2022), decomposing questions into sub-questions in a least-to-most fashion (Zhou et al., 2022), zero-shot prompting LLMs with simply one sentence (Kojima et al., 2022), writing programs to solve procedural tasks (Gao et al., 2022; Chen et al., 2022). Despite generating solutions in single forward pass, one line of work employs multiple reasoning results and ensembles them by majority vote (Wang et al., 2022), and stepwise verifier (Li et al., 2022). Additionally, Tree-of-Thoughts (Yao et al., 2023) deliberately explores multiple reasoning paths and searches over a tree-structured reasoning states. Imani et al. (2023) propose to vote over multiple solutions generated with algebraic and program prompts. One concurrent work (Zhao et al., 2023) considers the difference of CoT and PoT and asks the LLM to choose one better reasoning rationale. In contrast to their work, XoT involves more reliable verification modules and switches methods when necessary.

## 2.2 Iterative Refinement

One stream of work is dedicated to iteratively enhancing LLMs by continuously reevaluating and refining outputs until the desired quality is achieved. Madaan et al. (2023) prompts the model to write feedback based on previously generated drafts and leverages the feedback to generate high-quality outputs. Similarly, Chen et al. (2023) iteratively debugs the code by utilizing external program execution results and code explanations generated by the model itself. In order to avoid repetitive mistakes, Shinn et al. (2023) builds a memory of previous errors, while Wang and Li (2023) collects all mistakes during the training phase to provide a global insight. When considering sources of hints to guide rethinking, Paul et al. (2023) focuses on intermediate reasoning steps, while Zheng et al. (2023) directly utilizes the previously generated answers. Qi et al. (2023) propose to emulate the divide-and-conquer fashion of human thinking strategy and involve self-questioning and recursive thinking processes in the problem solving framework. Although these approaches contribute to improving the reasoning quality of LLMs, they are limited in retrying without looking around for other possible thoughts. In contrast, our proposed method aims to explore alternative solutions, and it is orthogonal

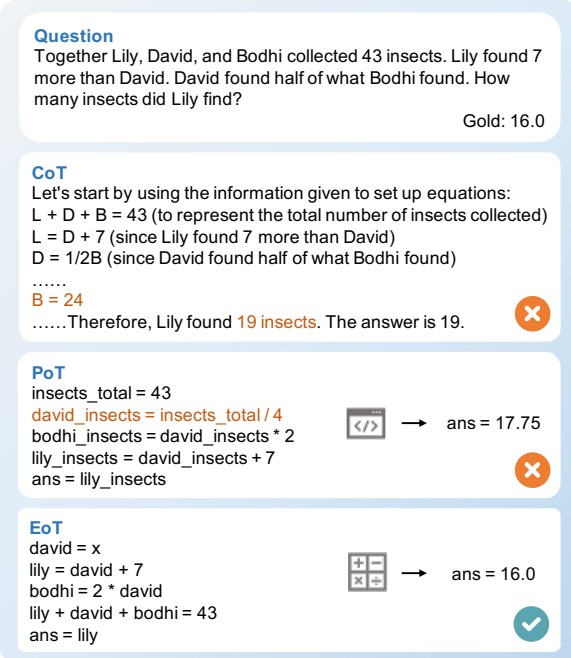

Figure 3: In particular cases where CoT and PoT fall short, EoT successfully solves the problem, which serves as a good complement.

to iterative refinement, as we have the flexibility to switch solutions when refining no longer leads to further improvement.

## 3 Preliminary

### 3.1 Prompting methods

For math reasoning tasks, we use three reasoning thoughts in this work, namely Chain-of-Thought (CoT), Program-of-Thought (PoT) and Equation-of-Thought (EoT). Despite the well-known strengths of CoT and PoT methods, our proposed EoT excels particularly in reasoning with unknown variables. For each problem, EoT attempts to model the questions as linear equations and involves unknown values in the description. A detailed formulation of EoT prompting can be found in Table 12 of Appendix C. As illustrated in Figure 3, while CoT correctly sets up the equations, it fails in accurately performing the calculations. PoT falls short in dealing with unknown variables, as Python requires that every variable is defined with a value. Assigning a value to an unknown variable (david_insects) hallucinates PoT to generate a misleading step (the highlighted line). In comparison, EoT manages to express the question context in straightforward equations and solves them with a deterministic equation solver.

## 3.2 Complementarity

Given a question $q$, we denote the correctness of the reasoning answers using each method as $\hat{R}_X(q)$, where $X \in \{CoT, PoT, EoT\}$ denotes the diverse reasoning methods. $\hat{R}_X(q) = \{0, 1\}$ represents whether the generated answer is correct according to the gold label. We define the accuracy under the oracle setting as:

$$ACC_{oracle} = \sum_q \hat{R}_{CoT}(q) \vee \hat{R}_{PoT}(q) \vee \hat{R}_{EoT}(q). \tag{1}$$

The oracle setting represents that the model has the potential for solving one given problem if any of the methods accurately generates the answer. It also implies that in cases where the generated answer does not match the gold answers, XoT will make further attempts using alternative methods to answer the question. Under oracle setting, the model can potentially achieve more than 10% gains on various datasets. In Figure 2, the bar at the bottom represents the highest performance achieved by employing a single method, followed by the optimal performance achieved through the use of two methods. The overall stacked bar shows the utilization of all three methods, which indicates the upper bound that can be reached through the combined collaboration of various methods.

## 4 XoT

Our goal is to develop a generalized problem solving framework that can automatically select the appropriate method for different problems and has the capability to switch among reasoning thoughts using both active and passive verification. We first describe the overall framework and introduce each module in detail.

### 4.1 Overall Framework

The overall pipeline is described in Algorithm 1. The inputs of our framework include a question $q$ and a predefined set of methods $M$. With the user input, XoT employs its three built-in modules to output the final solution, namely planning module $P$, reasoning module $R$ and verification module $V$.

These three modules collaborate in an iterative manner. Suppose at iteration $t$, the planning module $P$ first chooses the most appropriate method available: $m_t = P(M)$. The chosen method is subsequently excluded from the set of methods. The reasoning module is then tasked to generate

---

**Algorithm 1** XoT Reasoning Algorithm

**Require:** input question $q$, method set $M$, planning module $P$, reasoning module $R$, verification module $V$
1: $t \leftarrow 0$
2: **while** $|M| > 0$ **do**
3:     $m_t \leftarrow P(M)$         $\triangleright$ Choose method
4:     $M \leftarrow M \setminus \{m_t\}$
5:     $y \leftarrow R_{m_t}(q)$
6:     **if** $V(y)$ **then**
7:         break         $\triangleright$ Verification passed
8:     **else**
9:         $t \leftarrow t + 1$   $\triangleright$ Continue next iteration
10:     **end if**
11: **end while**
12: **return** $y$         $\triangleright$ Return the solution

---

one solution $y$ using the proposed method $m_t$. Following this, the verification module evaluates the solution by rethinking the answer within the given conditions. If the answer successfully passes the verification, we proceed to return the current solution. Otherwise, XoT will move forward to the next iteration. Every module is implemented with a LLM through inference under few-shot setting. We will elaborate each module with details.

### 4.2 Planning and Reasoning

The planning module is responsible for selecting the appropriate method at the beginning of each round of iteration. Recent work shows the necessity to equip reasoning framework with the ability to plan ahead (Lu et al., 2023). As elaborated in Section 3, it is evident that each method possesses distinct strengths. Our intuition is to consistently initiate the process with the optimal method to enhance reasoning efficiency.

The reasoning module performs few-shot reasoning with the planned prompting method. Each round of reasoning operates independently, meaning that subsequent iterations do not rely on the failed reasoning attempts of previous iterations.

### 4.3 Verification module

The verification module assesses the effectiveness of the reasoning solution through two approaches: passive verification and active verification.

When solutions involve offloading computation to external tools, the execution results naturally serve as a *passive verification*. Any occurrence of errors or exceptions during the execution directly

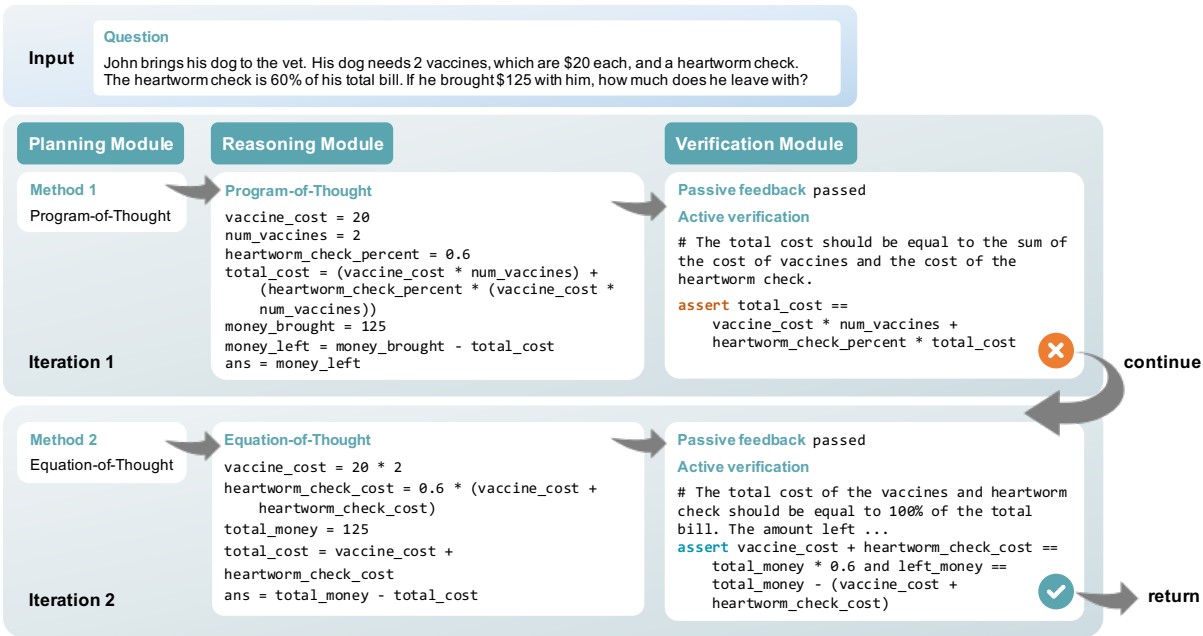

Figure 4: Overview of XoT. Following the suggestion of the planning module, XoT first reasons with PoT. However, the generated answer fails in the verification module. In the second iteration, the selected method is EoT. The reasoning module successfully generates the solution that passes the verification.

| Dataset | # Data | # Steps |
|---|---|---|
| GSM8K (Cobbe et al., 2021) | 1,319 | 3.25 |
| SVAMP (Patel et al., 2021) | 1,000 | 1.24 |
| AQuA (Ling et al., 2017) | 253 | $\geq 3^\star$ |
| Algebra (He-Yueya et al., 2023) | 222 | $\geq 2^\star$ |
| GSM-hard (Gao et al., 2022) | 1,313 | 3.25 |
| MATH (Hendrycks et al., 2021) | 5,000 | $\geq 3^\star$ |
| AddSub (Hosseini et al., 2014) | 395 | 1 |
| SingleOP (Roy et al., 2015) | 562 | 1 |
| SingleEQ (Koncel-Kedziorski et al., 2015) | 508 | 1.31 |
| MultiArith (Roy and Roth, 2015) | 600 | 2 |

Table 1: Statistics of the datasets we used. # Steps denotes the average number of reasoning steps in the gold answers. $\star$ indicates a rough estimate due to the inconsistent rationale formats.

results in a failure in the verification process. Solutions that pass the passive verification stage then proceed to active verification.

In the case of *active verification*, the module rethinks the answer within the context of the given question. It first acquires all intermediate values associated with each variable mentioned in the solution. These values are computed by external executors. We intentionally exclude the reasoning process (expressions) leading to the results to prevent the verification module from emulating the solution's thinking process. With the intermediate results and final answer in hand, the module is expected to recheck whether the answer satisfies the conditions specified in the question. The desired

format for this evaluation is an assertion statement, as shown in Figure 4. This assertion is subsequently combined with the original solution for external tools to execute. If no issues arise during this execution phase, it means the solution successfully passes the verification. A detailed illustration of the prompts we use can be found in Appendix C. The verification module is specially designed for PoT and EoT as the intermediate values can be easily obtained. We leave the exploration of a more effective verification for CoT as future work.

## 5 Experiments

### 5.1 Experimental Setting

**Datasets** Our experiments are conducted on a comprehensive set of 10 math reasoning datasets, encompassing various challenging math reasoning scenarios. Some widely used datasets include GSM8K, SVAMP, AQuA, MATH and MAWPS (AddSub, SingleOP, SingleEQ, Multi-Arith) (Koncel-Kedziorski et al., 2016). Besides, we also incorporate several recently introduced datasets, namely Algebra, GSM-hard. Algebra comprises a collection of solely algebraic word problems that can be resolved through the use of equations. To increase the complexity of calculations, GSM-hard replaced small numerical values with larger ones. The details of the statistics of the datasets can be found in Table 1.

**Model** We query OpenAI API for experiments[2]. Specifically we use `gpt-3.5-turbo` as the inference engine. If not further explained, we manually construct the prompts with 8 examples sampled from the training set. For CoT and PoT, we directly use the examples released by published paper (Fu et al., 2022; Gao et al., 2022; Chen et al., 2022). For model generation strategy, we employ greedy decoding in all runs. Due to the non-deterministic APIs, we report the average performance and the standard deviation across 3 runs. We also evaluate XoT with various base models in Appendix A.2.

## 5.2 Main Results

The main results are shown in Table 2. We consider three prompting methods as baselines, namely CoT, PoT and EoT. On average, XoT achieves a significant improvement of 5.49% across the datasets. For MATH dataset, we show the breakdown results of different question subtopics in Table 3. We also represent the performance enhancement over the strongest baseline as $\Delta$. As questions in MATH are too complex for equation systems to solve, we only consider CoT and PoT with passive verification. Specifically, on the AQuA dataset, which consists of multiple-choice questions, we observe that PoT or EoT often fails to generate a valid answer due to the diverse answer formats. Across the three runs, 24.4% of the PoT answers and 30.3% of the EoT answers cannot be executed. Therefore, applying passive verification is adequate to ensure the explortion of other method options. When post processing the generated results, we further enforce a restriction that the model cannot make a random guess if it fails to extract an answer from the generated output. Such instances should be proceeded to the next iteration to guarantee a fair evaluation of the performance.

Notably, we observe that the enhancements are more pronounced for the challenging datasets compared to the easier ones. Difficult datasets usually contain longer questions and more than 3 reasoning steps while easier datasets such as SingleEQ require only one equation to solve the problem. We find that the improvement directly correlates with the complementary nature of the three methods employed across different datasets. On easier datasets, each method performs well individually, resulting in only minor complementarity. Figure 5 reveals that XoT demonstrates superior performance on

---

[2]https://openai.com

---

datasets that exhibit stronger enhancement under oracle setting. The bars in the figure represent the improvement under XoT, while the line indicates the upper bound of the improvement under oracle setting. The comparison indicates that MultiArith and SingleEQ allow minimal room for improvement, therefore the overall XoT performance is negatively impacted by the accumulated errors introduced by the verification module.

Additionally, we conduct experiments on logical reasoning task to evaluate the generalisability of XoT. Details can be found in Appendix A.1.

## 6 Analysis

In this section, we first analyze the effectiveness and necessity of each module within XoT. Then we provide comparison with majority voting and describe how model's self refinement can be integrated in our framework.

### 6.1 Ablation Study

**Planning** The planning module decides which method to attempt at the beginning of each iteration. We are curious about how well it performs in selecting the most suitable method among the available options. The planning module is expected to select from PoT and EoT at the beginning because these two methods can be verified with both active and passive verification. To demonstrate the necessity of the planning module, we conduct an experiment in which XoT is asked to execute each method in a predefined order. Whether to switch the method is still determined by the verification module. We break down the performance of each dataset with respect to different combinations of methods in Table 4.

Our findings align with two design ethos of the planning module. First, it demonstrates **robustness** across different datasets. While specific combinations excel at different datasets, XoT equipped with the planning module outperforms all other predetermined combinations on average. For instance, on GSM-hard, the combination of PoT and EoT achieves the best performance, which highlights the importance of leveraging external tools to handle calculation involving large numbers. Additionally, on SingleEQ and MultiArith where XoT fails to offer improvement, the combination of two methods proves to be efficient, surpassing the single method baselines. With the inclusion of the planning module, XoT can dynamically adjust the

| | GSM8K | SVAMP | AQuA$^\star$ | Algebra | GSM-hard | AddSub | SingleOP | SingleEQ | MultiArith | Average |
|---|---|---|---|---|---|---|---|---|---|---|
| CoT | $80.2_{0.2}$ | $79.5_{0.6}$ | $55.1_{1.0}$ | $81.5_{0.8}$ | $42.4_{0.1}$ | $88.4_{0.3}$ | $93.4_{0.3}$ | $94.3_{0.1}$ | $\mathbf{97.5}_{0.3}$ | 79.14 |
| PoT | $77.2_{0.3}$ | $79.5_{0.3}$ | $49.2_{1.0}$ | $62.5_{0.7}$ | $61.8_{0.4}$ | $88.4_{0.2}$ | $93.4_{0.4}$ | $\mathbf{98.1}_{0.1}$ | $97.2_{0.0}$ | 78.59 |
| EoT | $63.8_{0.4}$ | $69.6_{0.7}$ | $46.7_{0.5}$ | $82.3_{0.5}$ | $53.8_{0.2}$ | $71.6_{1.0}$ | $75.4_{0.4}$ | $85.8_{0.8}$ | $78.6_{0.6}$ | 69.73 |
| XoT | $\mathbf{83.3}_{0.5}$ | $\mathbf{83.6}_{0.6}$ | $\mathbf{61.7}_{0.6}$ | $\mathbf{89.9}_{0.3}$ | $\mathbf{63.4}_{0.5}$ | $\mathbf{90.5}_{0.4}$ | $\mathbf{94.3}_{0.3}$ | $97.7_{0.1}$ | $97.3_{0.3}$ | $\mathbf{84.63}$ |
| oracle | $92.5_{0.2}$ | $92.7_{0.3}$ | $77.0_{1.4}$ | $95.5_{0.5}$ | $74.3_{0.4}$ | $93.9_{0.3}$ | $97.5_{0.0}$ | $99.1_{0.1}$ | $99.3_{0.0}$ | 91.31 |
| Δ | +3.1 | +4.1 | +6.6 | +7.6 | +1.6 | +2.1 | +0.9 | -0.4 | -0.2 | +5.49 |

Table 2: Main experiment results across various math reasoning datasets. Under oracle setting, XoT switches the method if the generated answer does not match the gold answers. $\star$ denotes we only use passive verification. Δ represents the improvement over the best performing baseline.

| | InterAlgebra | Precalculus | Geometry | NumTheory | Probability | PreAlgebra | Algebra | Overall |
|---|---|---|---|---|---|---|---|---|
| CoT | $17.8_{0.4}$ | $20.3_{0.4}$ | $24.4_{0.4}$ | $32.2_{1.0}$ | $30.4_{0.6}$ | $56.6_{0.4}$ | $49.7_{0.4}$ | $35.77_{0.4}$ |
| PoT | $14.4_{0.1}$ | $15.5_{0.1}$ | $8.8_{0.3}$ | $31.2_{0.7}$ | $19.6_{0.2}$ | $36.5_{0.2}$ | $38.2_{0.1}$ | $25.79_{0.1}$ |
| XoT | $\mathbf{25.1}_{0.1}$ | $\mathbf{26.0}_{0.3}$ | $\mathbf{25.3}_{0.7}$ | $\mathbf{48.1}_{0.6}$ | $\mathbf{36.1}_{0.4}$ | $\mathbf{62.0}_{0.2}$ | $\mathbf{57.3}_{0.4}$ | $\mathbf{42.81}_{0.0}$ |
| oracle | $28.1_{0.2}$ | $31.2_{0.1}$ | $27.6_{0.4}$ | $50.5_{0.1}$ | $39.0_{0.7}$ | $68.0_{0.4}$ | $64.1_{0.4}$ | $47.35_{0.2}$ |
| Δ | +7.3 | +5.7 | +0.9 | +15.9 | +5.7 | +5.4 | +7.6 | +7.04 |

Table 3: Experiment results on MATH dataset. We only employ two methods and passive verification on MATH.

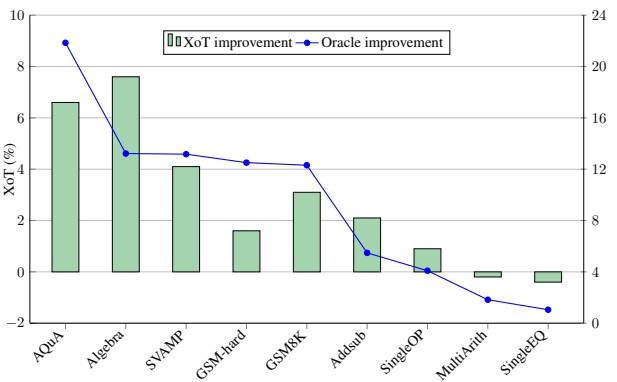

Figure 5: The correlation between oracle performance and final improvement. A higher oracle gain allows more room for XoT to improve.

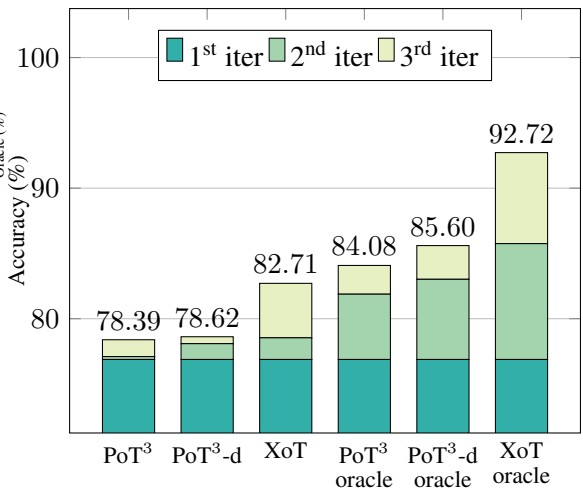

Figure 6: Repeatedly exploiting the same method (PoT$^3$) results in limited complementarity compared to XoT with three methods. PoT$^3$-d denotes we use different few-shot examples in three iterations.

execution order based on different questions, which ensures a more consistent and robust performance.

Second, the planning module enhances **efficiency**, facilitating XoT to reach the final answer in fewer iterations by always starting from the most possible method. To illustrate, on GSM8K, XoT needs 1.46 iterations on average in comparison with 1.58 iterations with the fixed EPC order (EoT->PoT->CoT, the best performing fixed order). Specifically, 68.8% of the questions are resolved in the first iteration with XoT, as opposed to 57.2% when employing the fixed EPC order.

**Reasoning** How important is it to try different methods instead of exclusively relying on a single method? To investigate this, we restrict the available method options to utilizing PoT only, denoted

as PoT$^3$. In other words, if the generated solution fails to pass the verification, it reconsiders its reasoning using the same prompting method instead of changing to another. The results are demonstrated in Figure 6. PoT$^3$ uses the same few-shot examples in three iterations while PoT$^3$-d uses differente examples randomly sampled from the training set. It is observed that under orcale setting, repetitive exploitation of a single method has limited complementarity of 84.08%, which is 8.64% less than XoT. As a result, the final performance reflects such a gap with PoT$^3$ of 78.39% and XoT of 82.71%.

| Methods | GSM8K | SVAMP | AQuA | Algebra | GSM-hard | AddSub | SingleOP | SingleEQ | MultiArith | Average |
|---|---|---|---|---|---|---|---|---|---|---|
| PE | $77.7_{0.3}$ | $80.7_{0.2}$ | $56.7_{1.0}$ | $81.7_{0.5}$ | $63.4_{0.3}$ | $89.6_{0.3}$ | $93.8_{0.3}$ | $98.0_{0.2}$ | $95.0_{0.2}$ | 81.85 |
| PC | $81.8_{0.2}$ | $82.7_{0.6}$ | $61.7_{1.5}$ | $83.6_{0.5}$ | $59.6_{0.4}$ | $90.4_{0.0}$ | $94.4_{0.2}$ | $\mathbf{98.3}_{0.1}$ | $\mathbf{97.8}_{0.2}$ | 83.36 |
| EP | $80.9_{0.4}$ | $80.8_{0.4}$ | $58.0_{0.6}$ | $83.8_{0.9}$ | $\mathbf{64.6}_{0.3}$ | $88.4_{0.4}$ | $94.1_{0.5}$ | $96.7_{0.0}$ | $\mathbf{97.8}_{0.2}$ | 82.80 |
| EC | $82.4_{0.5}$ | $81.4_{0.6}$ | $60.0_{0.6}$ | $\mathbf{92.0}_{0.3}$ | $56.2_{0.4}$ | $87.3_{0.4}$ | $93.7_{0.2}$ | $95.1_{0.1}$ | $97.3_{0.2}$ | 82.82 |
| EPC | $82.6_{0.5}$ | $82.6_{0.6}$ | $\mathbf{63.1}_{1.0}$ | $89.9_{0.3}$ | $63.1_{0.4}$ | $88.7_{0.6}$ | $\mathbf{94.5}_{0.3}$ | $96.7_{0.0}$ | $97.5_{0.0}$ | 84.29 |
| PEC | $82.6_{0.4}$ | $83.1_{0.5}$ | $61.8_{1.0}$ | $85.3_{0.5}$ | $63.3_{0.3}$ | $90.1_{0.3}$ | $94.4_{0.3}$ | $98.2_{0.1}$ | $97.4_{0.3}$ | 84.02 |
| XoT | $\mathbf{83.3}_{0.5}$ | $\mathbf{83.6}_{0.6}$ | $61.7_{0.6}$ | $89.9_{0.3}$ | $63.4_{0.5}$ | $\mathbf{90.5}_{0.4}$ | $94.3_{0.3}$ | $97.7_{0.1}$ | $97.3_{0.3}$ | $\mathbf{84.63}$ |

Table 4: Results across different datasets without the planning module. We manually define the execution sequence, denoted as the combination of the first letter in each method. For example, 'PEC' indicates PoT-EoT-CoT.

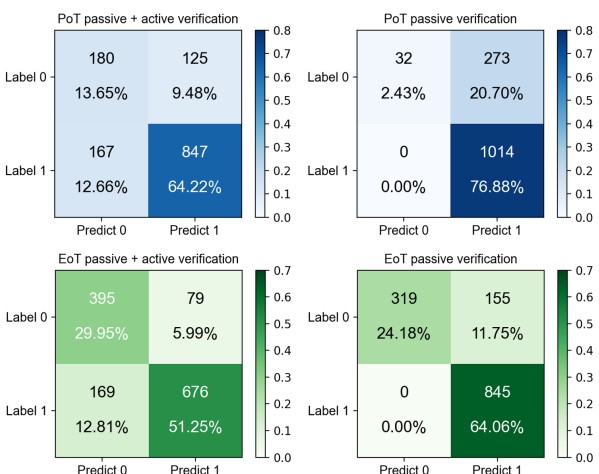

Figure 7: Comparison of passive and active verifications. The blue and green matrices represent verifications for PoT and EoT respectively.

| | active | ACC | FPR↓ | FNR↓ | XoT |
|---|---|---|---|---|---|
| PoT | ✗ | 79.3 | 89.5 | 0.0 | 80.4 |
| EoT | ✗ | 88.3 | 32.7 | 0.0 | |
| PoT | ✓ | 77.9 | 41.0 | 16.5 | 82.7 |
| EoT | ✓ | 81.2 | 16.7 | 20.0 | |

Table 5: Ablation results of different verification methods on GSM8K. Employing active verification significantly reduces false positive rate and results in a notable improvement in the overall XoT performance.

This suggests the necessity of employing various reasoning methods in our framework.

**Verification** The verification module facilitates seamless switching between iterations. We here explore how helpful the active and passive verifications are. Figure 7 illustrates the performance comparison when considering different verification aspects. If we solely depend on passive verification, only 2.43% of the PoT results and 24.18% of the EoT results are deemed "incorrect" and subsequently advanced to the next iteration. However,

| | GSM8K | SVAMP |
|---|---|---|
| PoT | $77.2_{0.3}$ | $79.5_{0.6}$ |
| EoT | $63.8_{0.4}$ | $69.6_{0.7}$ |
| XoT (only PE) | $79.4_{0.7}$ | $81.3_{0.3}$ |
| XoT (w/o verification) | $74.5_{1.4}$ | $79.2_{0.4}$ |

Table 6: Ablation results of excluding the entire verification module on GSM8K and SVAMP. XoT (only PE) is equipped with the verification module. The lack of this module compromises its ability for iterative method-switching, resulting in diminished performance.

such a simplistic verification approach yields an alarmingly high false positive rate of 89.5% and 41.0%, as shown in Table 5. This drawback is particularly critical as our XoT's essence lies in the ability to adaptively switch methods, and a high false positive rate restricts the model's ability to explore alternative method options. By additionally incorporating active verification, despite a slight compromise in accuracy, the false positive rate is substantially reduced by 56.8% and 24.3%. We also note that this approach inevitably leads to an increase in the false negative rate. However, this is a minor drawback as the subsequent method options still have chances to get it correct. Consequently, employing active verification offers 2.3% gains to the overall XoT performance.

Additionally, we explore the necessity of the iterative nature of XoT by removing the entire verification module. In this scenario, we only reason once with the most suitable method suggested by the planning module. The results are presented in Table 6. As our planning module mainly chooses the method from PoT or EoT, we here restrict the available methods to PoT and EoT only in XoT framework, which is denoted as 'XoT (only PE)'. By removing the verification module, the framework, denoted by 'XoT (w/o verification)' is no more capable of rechecking the answer thus cannot

| | GSM8K | SVAMP | AQuA | Algebra | GSM-hard | AddSub | SingleOP | SingleEQ | MultiArith | Average | #Tokens |
|---|---|---|---|---|---|---|---|---|---|---|---|
| XoT | $83.3_{0.5}$ | $83.6_{0.6}$ | $61.7_{0.6}$ | $89.9_{0.3}$ | $63.4_{0.5}$ | $90.5_{0.4}$ | $94.3_{0.3}$ | $97.7_{0.1}$ | $97.3_{0.3}$ | 84.63 | 4.5k |
| Vote | $82.4_{0.2}$ | $84.7_{0.8}$ | $55.6_{1.9}$ | $79.7_{0.5}$ | $61.3_{1.1}$ | $89.4_{0.4}$ | $94.4_{0.1}$ | $97.2_{0.1}$ | $98.5_{0.2}$ | 82.59 | 5.4k |

Table 7: Comparison between XoT and Majority Voting. XoT outperforms the majority vote approach in a more efficient manner, yielding an average gain of 2.04 with a reduction of 16.7% in token count. #Tokens denotes the average number of tokens consumed for one case (including prompts, question and response).

| | ACC | ACC + refine |
|---|---|---|
| CoT | 80.4 | 81.7 |
| PoT | 76.9 | 76.9 |
| EoT | 64.1 | 66.5 |
| XoT | 82.7 | 84.5 |

Table 8: Results of adding self-refinement within reasoning module on GSM8K test set.

perform iterative attempts to switch methods. This leads to a performance degradation of 4.9% and 2.9% on GSM8K and SVAMP respectively.

## 6.2 Comparison with Majority Voting

We additionally conduct experiments involving the majority vote of three distinct methods. The vote is based on three answers generated by three methods (one answer per method). As shown in Table 7, taking the majority vote of the three methods achieves 82.59 on average, while XoT achieves better performance at 84.63. Additionally, we observe that the majority vote fails on datasets containing questions that align exceptionally well with a specific method. Specifically, the majority vote achieves 79.73 on Algebra, while XoT achieves 89.94.

The majority vote needs to execute all three methods to reach an answer, while XoT will stop when the answer passes the verification. We calculate the total token count as $\#\texttt{total\_token} = \#\texttt{input\_token} + \#\texttt{output\_token} * 2$, according to OpenAI's pricing policy[3]. As shown from the table, XoT is able to achieve higher performance with a lower budget, exhibiting a reduction of 16.7% in expenses. The token count includes all the in-context examples used and is averaged across the number of the total questions in 9 datasets.

## 6.3 Self-refinement

The design principle underlying XoT is its adaptable capability to switch methods, allowing for smooth integration with research aimed at improving individual methods. The line of iterative refinement methods enhances the model performance

by asking the model to rethink on its previous response, serving as a good alternative for the reasoning module in XoT. Specifically, before moving on to another method at each iteration, we allow the model to first make self refinement on its current approach, making the best use of current method.

Inspired by previous work (Madaan et al., 2023), after reasoning with one method for the first time, we require the model to analyze its response line-by-line and summarize several advice to mitigate the potential errors. Then, the model answers the question for a second time in the same method, with the summarized advice as a hint. After that, we verify the results produced by the second round and determine whether to switch to another method.

To achieve the iterative refinement in CoT, we follow Zheng et al. (2023) to progressively hint the model with the answers generated before. For PoT and EoT, we follow the released self-refinement prompts from Madaan et al. (2023). The results are shown in Table 8. We only allow the model to think twice using each prompting method. Though adding only one round of refinement yields marginal improvement within each single method, their collaboration contributes to a more significant improvement under XoT framework.

## 7 Conclusion

We propose XoT, an integrated problem solving framework that utilizes diverse reasoning thoughts to prompt LLMs. XoT integrates planning, reasoning and verification into a unified framework, enabling the model to explore multiple methods based on the active and passive verification of the solutions. We conduct extensive experiments on 10 math reasoning datasets to thoroughly evaluate the advantages of each module and showcase the efficacy of our proposed approach. Further results also show that the design ethos of XoT can generalize to logic reasoning domain. We consider its generalisation to more diverse tasks as a compelling avenue for future exploration.

---

[3] https://openai.com/pricing

## Limitations

We acknowledge that our approach falls short on easier and more straightforward datasets where different methods exhibit limited complementary relations. Our current approach relies on the availability of diverse prompting methods for reasoning tasks. Further research is required to explore new problem solving methods for general reasoning tasks. Moreover, we observe that our method works better on larger base models. Although different reasoning methods do exhibit notable complementarity on smaller models, the inherent potential is not yet fully unleashed in current XoT design.

## Ethics Statement

The data used in our work all comes from public dataset, and our proposed method can be further integrated with other methods. Our work is conformant to ACL Ethics Policy.

## Acknowledgements

We would like to thank the anonymous reviewers for their valuable suggestions and feedback. This work was supported by the National Natural Science Foundation of China (No. 62236004 and No. 62022027).

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

# A Further Analysis

## A.1 Generalisation to logical domain

We analyze the generalisability of XoT framework to logical reasoning domain. One recent work (Yao et al., 2023) proposed LogicLM to solve logical reasoning questions using First Order Logic expressions and executed them in external symbolic reasoners. Following LogicLM, we design similar formal language expressions to represent First Order Logic and conduct experiments on FOLIO (Han et al., 2022), an expert-written, logically complex and diverse dataset for natural language reasoning. Our findings in Table 9 suggest that different methods in logical domain also show strong complementarity, achieving 77.45% under oracle setting. After involving the verification module, XoT performs at 62.75% on the validation set of FOLIO. These results underscore the applicability of XoT as a general problem solving framework.

| Method | FOLIO ACC |
|--------|-----------|
| CoT | 58.82 |
| FOL | 42.65 |
| Oracle | 77.45 |
| XoT | **62.75** |

Table 9: XoT performance on logical reasoning task FOLIO validation set. Normal text reasoning and formal language FOL are complement to each other under oracle setting and XoT framework.

## A.2 Experiments on other models

We further assess the performance of XoT across various base models, such as Llama-2 series (Touvron et al., 2023b). The results are shown in Table 10, and we illustrate the performance scaling curve in Figure 8. With less capable models, different prompting methods still demonstrate strong complementarity under oracle setting. Our observations suggest that smaller models tend to yield suboptimal results, likely due to the unbalanced performance across different reasoning approaches

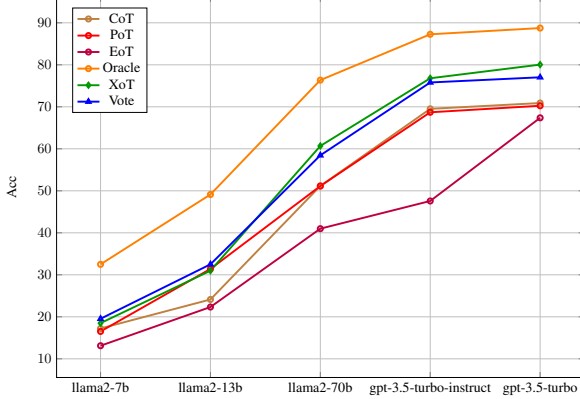

Figure 8: Performance scaling curve on different base models. The performance is averaged across the four datasets shown in Table 10.

and the models' limited capability for active verification. This limitation inhibits the model's ability to timely switch between methods. However, as the model's size increases, XoT consistently shows its strength across the datasets.

### A.3 Proportion of XoT

Figure 9 illustrates the proportion of different methods that XoT selects as the final answers. On GSM8K, 56.7% questions end up being solved with PoT, while 28.3% are tackled by EoT. The remaining 15% is left for CoT to solve.

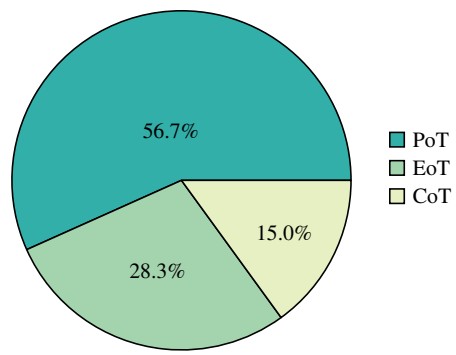

Figure 9: The proportion of different methods that XoT finally chooses as the answer on GSM8K.

### B  XoT with self refinement

We here offer the details of how we combine iterative self-refinement with XoT framework. As shown in Figure 10, the self refinement process can be integrated in the reasoning module, where the dashed line indicates rethinking using the same method. When the desired number of self refinement iterations is reached, the generated solutions will proceed to the verification module. Then the

verification will determine whether to use the current solution or change to another method.

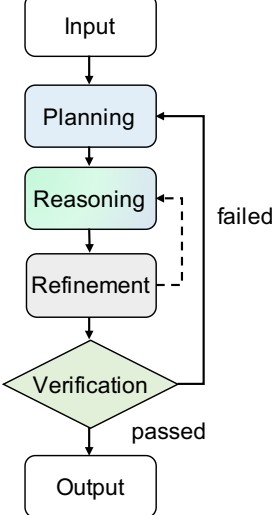

Figure 10: Self refinement can be integrated in the XoT framework. The dashed block indicates the reasoning module with the inclusion self refinement. Within each self refinement process, the model repeatedly exploits the same method.

### C  Examples

In this section, we show the input and output examples of each module in XoT. Full prompts are available in public Github repository: `https://github.com/tengxiaoliu/XoT`. For EoT, we use sympy [4] library to solve the linear equations.

---

[4]`https://www.sympy.org/`

| | Method | GSM8K | GSM-hard | Algebra | SVAMP | Average |
|---|---|---|---|---|---|---|
| | CoT | 14.3 | 3.8 | 17.6 | 33.3 | 17.23 |
| | PoT | 10.2 | 8.3 | 14.9 | 32.8 | 16.52 |
| | EoT | 8.0 | 5.0 | 16.2 | 23.2 | 13.12 |
| LLaMA2-7B | Oracle | 24.6 | 14.6 | 32.0 | 58.9 | 32.50 |
| | XoT | 13.5 | 5.9 | 19.4 | 35.2 | 18.50 |
| | Vote | 13.9 | 7.0 | 19.8 | 37.4 | 19.54 |
| | CoT | 27.5 | 8.5 | 20.3 | 40.3 | 24.13 |
| | PoT | 26.4 | 22.7 | 26.1 | 50.4 | 31.40 |
| | EoT | 14.5 | 12.1 | 32.9 | 29.9 | 22.33 |
| LLaMA2-13B | Oracle | 46.1 | 32.2 | 49.6 | 68.7 | 49.13 |
| | XoT | 30.2 | 17.4 | 30.2 | 46.2 | 30.98 |
| | Vote | 28.6 | 18.9 | 31.5 | 51.0 | 32.50 |
| | CoT | 59.1 | 27.1 | 43.2 | 75.2 | 51.16 |
| | PoT | 52.3 | 43.1 | 35.6 | 73.6 | 51.14 |
| | EoT | 31.0 | 25.0 | 63.5 | 44.4 | 40.99 |
| LLaMA2-70B | Oracle | 78.3 | 59.3 | 78.8 | 89.0 | 76.36 |
| | XoT | 57.9 | 43.8 | 64.0 | 77.0 | 60.68 |
| | Vote | 58.6 | 38.9 | 56.3 | 80.0 | 58.45 |
| | CoT | 77.9 | 44.4 | 82.0 | 73.8 | 69.52 |
| | PoT | 72.5 | 59.5 | 64.4 | 78.4 | 68.70 |
| | EoT | 41.3 | 36.1 | 65.3 | 47.6 | 47.58 |
| gpt-3.5-turbo-instruct | Oracle | 90.1 | 73.7 | 95.1 | 90.2 | 87.27 |
| | XoT | 77.8 | 58.8 | 90.5 | 80.1 | 76.80 |
| | Vote | 77.9 | 58.2 | 85.6 | 81.5 | 75.80 |

Table 10: XoT performances with various base models.

---

**Input:**

You need to choose the best method for the given question. Each method has its own strength. The methods are described as follows:

- Python Program: This method generates a Python program that can solve the given question. It takes in the question and possible context and produces a program. Normally, we consider using this method when the questions and contexts involve forward reasoning, such as arithmetic operations over multiple numbers, or when the questions involve complex logical operations, such as "if-else" statements.

- System of linear equations: This method builds a math model and generates a system of linear equations that contains the answer as an unknown variable. Normally, we consider using this method when the questions and contexts involve an unknown variable that must be used to build an equation, especially when the question can be better modeled with abstract mathematical declarations, or when the unknown variable appears at the beginning of the questions and needs backward reasoning to solve.

Below are some examples that choose the most appropriate method for the math word problems.

{Examples}

Question: Alyssa, Keely, and Kendall ordered 100 chicken nuggets from a fast-food restaurant. Keely and Kendall each ate twice as many as Alyssa. How many did Alyssa eat?

Method:

---

**Output:**

System of linear equations

---

Table 11: Example of planning module.

| Input: |
| --- |
| {Examples} |
| Question: Alyssa, Keely, and Kendall ordered 100 chicken nuggets from a fast-food restaurant. Keely and Kendall each ate twice as many as Alyssa. How many did Alyssa eat? |
| System of linear equations: (Do not simplify) |

| Output: |
| --- |
| % Assume Alyssa ate x chicken nuggets: |
| alyssa = x |
| % Alyssa, Keely, and Kendall ordered 100 chicken nuggets from a fast-food restaurant: |
| total = 100 |
| alyssa + keely + kendall = total |
| % Keely and Kendall each ate twice as many as Alyssa: |
| keely = 2 * alyssa |
| kendall = 2 * alyssa |
| % How many did Alyssa eat? The answer is alyssa: |
| ans = alyssa |

Table 12: Example of reasoning in EoT.

| Program: |
| --- |
| nuggets_total = 100 |
| nuggets_keely = 2 * nuggets_alyssa |
| nuggets_kendall = 2 * nuggets_alyssa |
| nuggets_alyssa = (nuggets_total - nuggets_keely - nuggets_kendall) / 5 |
| ans = nuggets_alyssa |

| Python Interpreter: |
| --- |
| NameError: name 'nuggets_alyssa' is not defined |

Table 13: Example of a failed passive verification on PoT. The Python interpreter throws an error, indicating the solution does not pass the passive verification.

| Input: |
| --- |
| {Examples} |
| Question: Alyssa, Keely, and Kendall ordered 100 chicken nuggets from a fast-food restaurant. Keely and Kendall each ate twice as many as Alyssa. How many did Alyssa eat? |
| alyssa = 20 |
| total = 100 |
| keely = 40 |
| kendall = 40 |
| # Assertion |

| Output: |
| --- |
| # The total number of chicken nuggets should be equal to the sum of the nuggets Keely, Kendall and Alyssa ate. The number of Keely and Kendall each ate should be twice as many as Alyssa. |
| assert total == alyssa + keely + kendall |
| assert keely == 2 * alyssa and kendall == 2 * alyssa |

Table 14: Example of an active verification on an EoT solution.

**Input:**

{Examples}

Question: Alyssa, Keely, and Kendall ordered 100 chicken nuggets from a fast-food restaurant. Keely and Kendall each ate twice as many as Alyssa. How many did Alyssa eat?

alyssa = 20
total = 100
keely = 40
kendall = 40
# Assertion

---

**Output:**

# The total number of chicken nuggets should be equal to the sum of the nuggets Keely, Kendall and Alyssa ate. The number of Keely and Kendall each ate should be twice as many as Alyssa.
assert total == alyssa + keely + kendall
assert keely == 2 * alyssa and kendall == 2 * alyssa

Table 15: Example of an active verification on an EoT solution.