# OpenReview forum: "Plan, Verify and Switch: Integrated Reasoning with Diverse X-of-Thoughts"
_EMNLP/2023/Conference — EMNLP 2023 Main_

### Official Review · Reviewer_wiK8 · 2023-08-03

**Soundness:** 3

**Excitement:**

3: Ambivalent: It has merits (e.g., it reports state-of-the-art results, the idea is nice), but there are key weaknesses (e.g., it describes incremental work), and it can significantly benefit from another round of revision. However, I won't object to accepting it if my co-reviewers champion it.

**Paper Topic And Main Contributions:**

This paper proposes a problem-solving framework for math reasoning tasks. Specifically, it iteratively selects one of the existing prompting methods and then executes it. At the same time, it actively checks the validity of generated answers using passive and active verification.

**Questions For The Authors:**

A. Some details of the planning method are missing. In your prompt, you introduce just PoT and EoT, but in Table 4 there is CoT. And it seems that EoT+PoT+CoT could achieve a much more desirable performance (84.29 v.s. 84.63). Can it be understood that simply combining several prompting methods will achieve good results? Why don't you try other orders, such as CoT+EoT+PoT or CoT+PoT+EoT?
B. It is obvious that combining various methods can bring improvement, but where this improvement comes from is still unknown. Is it because CoT, PoT, and EoT are good at different types of mathematical problems, or just because the model is allowed to think more times? Increased analysis in this aspect will help.

**Reasons To Accept:**

1. This paper proposes an automatic problem-solving framework XoT, utilizing the complementarity of different reasoning thoughts.
2. This paper incorporates passive and active verification to facilitate the framework.

**Reasons To Reject:**

1. The main motivation of this paper comes from the combination of various prompting methods, while from experimental results it seems that simply combining several prompting methods could already achieve promising results (84.29 v.s. 84.63). Therefore, the novelty is somewhat lacking.
2. CoT, PoT, and other prompting methods are applicable to various reasoning scenarios, while this paper only conducts experiments on the math reasoning datasets, thus the generalizability of the methods remains unknown.
3. Experiments conducted only use the gpt-3.5-turbo model, hence the inclusion of more LLMs (e.g., davinci from Openai, and other open-source models such as Llama & Alpaca) would be desirable.

**Reproducibility:**

3: Could reproduce the results with some difficulty. The settings of parameters are underspecified or subjectively determined; the training/evaluation data are not widely available.

**Reviewer Confidence:**

4: Quite sure. I tried to check the important points carefully. It's unlikely, though conceivable, that I missed something that should affect my ratings.

---

> ### Author Rebuttal · Authors · 2023-08-29
>
> Thank you for your valuable and constructive feedback. We provide specific responses and clarifications as follows.
>
> > Reason 1 & Question A: Clarification on the ablation results of the planning module
>
> - _"Can it be understood that simply combining several prompting methods will achieve good results?"_
>
> Sorry for the confusion on the interpretation of 84.29 v.s. 84.63. Regarding the result, we would like to clarify that 84.29 (EoT->PoT->CoT) represents the ablation result of the planning module. In other words, 84.29 is achieved with both reasoning and verification modules in XoT, which demonstrates the effectiveness of these designs in our method.
>
> The main difference between 84.29 and 84.63 is the planning module, which can dynamically determine the execution order of different prompting methods. Specifically, the design principle of the planning module is for **robustness** and **efficiency**, instead of a huge performance boost.
>
> (1) For robustness, although different fixed method orders demonstrate their respective advantages across various datasets, none of them excel across all datasets (Table 4). The planning module can dynamically adjust the execution order based on different questions, which ensures a more consistent and robust performance on average.
>
> (2) For efficiency analysis, the planning module facilitates XoT to reach the final answer in fewer iterations by always starting from the most possible method. Specifically on GSM8K, XoT needs 1.46 iterations on average in comparison with 1.58 iterations with the fixed EPC order (the best performing fixed order). We will add the clarification into the analysis section accordingly.
>
> - _“Why don’t you try other orders?”_
>
> The initial decision to not select CoT as the first method is mainly due to the absence of a reliable verification module to specify the correctness of CoT answers. We provided the intuition of the verification module design in Line 306-310. In detail, the planning module mainly selects from PoT and EoT as these two methods incorporate external executors that can provide intermediate values and deterministic execution results, which will be used by both passive and active verification.
>
> - _"the novelty is somewhat lacking"_
>
> We appreciate the reviewer's attention to the novelty of our work. While XoT builds on the foundation of various independent reasoning methods, our work provides initial empirical evidence that different prompting methods have complementarity rather than exclusive competencies. The main highlight of our work is to present a new problem solving framework that involves different methods to work together, which has not been explored comprehensively before.
>
> Additionally, we add the effectiveness and efficiency analysis between XoT and the traditional ensemble method majority vote. Our findings indicate that XoT outperforms the majority vote approach in a more efficient manner, yielding a gain of 2.04 with a reduction of 16.7% in token count. A detailed description of the results is available in the response [Reason 3 & Question 5] for Reviewer JpnC.
>
> > Reason 2: Generalizability to other domains
>
> Due to space constraints, this paper mainly employs math reasoning as a problem solving scenario. XoT represents a new problem solving framework which is not limited to the reasoning methods described in this paper. We noticed a recent work (Pan et al., 2023) proposed to solve logical reasoning questions using First Order Logic expressions and executed them in external symbolic reasoners. We conducted initial analysis on the complementarity of this method (denoted as FOL) and normal CoT. As shown in the table below, these two methods are well complement to each other, performing at 70.44 under the oracle setting. This finding underscores the potential applicability of XoT within the domain of logical reasoning. We leave the further exploration as a promising avenue for future investigation.
>
> | Method | Acc on FOLIO (Han et al., 2022)  |
> |:------:|:----------------------------:|
> | CoT    |            57.84             |
> | FOL    |            45.32             |
> | Oracle |            70.44             |
>
> Caption: FOLIO is an expert-written, logically complex and diverse dataset for natural language reasoning.
>
> > Reason 3: Experiments on other models
>
> We thank the reviewer for this suggestion, which we believe helps strengthen our results. Per the reviewer’s request, we conduct additional experiments on GSM8K with both text-davinci-003 and text-davinci-002, which are less performing models than gpt-3.5-turbo (text-davinci-002 is trained with supervised fine-tuning w/o RL). The results are shown below:
>
> | Method | text-davinci-003 | text-davinci-002 |
> |--------|------------------|------------------|
> | CoT    | 67.63            | 56.86            |
> | PoT    | 66.41            | 59.82            |
> | EoT    | 50.42            | 42.76            |
> | XoT    | 71.11 (+3.48)    | 64.29 (+4.47)    |
> | Oracle | 84.91            | 78.85            |
>
> With less capable models, different prompting methods still demonstrate strong complementarity under oracle setting. Specifically, XoT is able to achieve consistent gains (+3.48,+4.47) with both models. In addition, we note that XoT is agnostic to the specific models, as long as the models exhibit the capability to perform CoT reasoning. The determinant influencing XoT performance resides in the complementary nature manifested between the dataset and the prompting methods.
>
> > Question B: where this improvement comes from is still unknown
>
> We acknowledge the importance to analyze the source of  XoT's improvement.
> The underlying reason for the improvement indeed aligns with your first viewpoint: different prompting methods are good at different types of questions and they are complementary to each other.
> We analyzed the complementarity of these methods in section 3.2 and Figure 2. Our oracle analysis provided empirical evidence that various methods have unique strengths on different problems. For instance, the question illustrated in Figure 3 is especially appropriate to be solved with EoT. We will emphasize the source of the improvement in the writing for a clearer presentation.
>
> We agree that thinking more times is beneficial for models to achieve better performance. Our empirical findings suggest that XoT could provide further gains over simply thinking more times. We conducted experiments on reasoning with a single method for the same number of iterations and provided comparison in section 6.1 (reasoning ablation) and Figure 6. Although the model is allowed to think multiple times with PoT (denoted as PoT^3), a single method results in limited complementarity (-8.64  than XoT) and overall gain (-4.32 than XoT), which indicates the importance of employing different prompting methods.
>
> We are grateful for your insight and constructive feedback, and we are committed to refining the paper to present a more comprehensive and accurate representation of our work. We would highly appreciate it if you could reassess our paper based on our response and let us know if you might find our work more positive.
>
> References:
> - Pan et al., 2023. LOGIC-LM: Empowering Large Language Models with Symbolic Solvers for Faithful Logical Reasoning.
> - Han et al., 2022. FOLIO: natural language reasoning with first-order logic.

---

### Official Review · Reviewer_hMke · 2023-08-03

**Soundness:** 4

**Excitement:**

3: Ambivalent: It has merits (e.g., it reports state-of-the-art results, the idea is nice), but there are key weaknesses (e.g., it describes incremental work), and it can significantly benefit from another round of revision. However, I won't object to accepting it if my co-reviewers champion it.

**Paper Topic And Main Contributions:**

The paper proposes a problem-solving framework called XoT that utilizes large language models (LLMs) with different prompting methods, such as Chain of Thought (CoT), Program of Thought (PoT), and Equation-of-Thought (EoT), to improve math reasoning tasks. XoT selects the most suitable method for each question and iteratively executes them, actively checking the validity of generated answers and incorporating feedback from external executors. The proposed approach is evaluated on 10 popular math reasoning datasets, demonstrating its effectiveness and the strengths of each module. XoT allows for dynamic switching among different prompting methods, providing a unified framework that integrates diverse reasoning thoughts collaboratively. The results suggest that this approach complements recent improvements in single reasoning methods and opens up new possibilities for math problem-solving.

**Questions For The Authors:**

1) How do you run the generated EoT program? I do not find any implementation details.

**Reasons To Accept:**

1) The paper proposes XoT, a unified framework that integrates diverse reasoning modules collaboratively with planning and verification modules to improve the reasoning ability of LLMs.
2) The paper further explores a new prompting method called Equation-of-Thought (EoT), inspired by Linear Algebra, which forms the math problem as a linear equation system, enabling a more direct approach to math reasoning.
3) Experimental results on 10 math datasets show XoT is better than any single prompting method.

**Reasons To Reject:**

1) The ensemble of three different prompting strategies leads to better performance is not surprising.
2) The planning module in XoT seems not to lead to significant gains compared with verification only. Adding efficiency analysis might be needed to further illustrate the benefits of the planning module.
3) Active verification seems to decrease the overall accuracy. A deep analysis of this part might be needed. The authors may also consider showing the results without any passive or active verification in the ablation study.

**Reproducibility:**

3: Could reproduce the results with some difficulty. The settings of parameters are underspecified or subjectively determined; the training/evaluation data are not widely available.

**Reviewer Confidence:**

5: Positive that my evaluation is correct. I read the paper very carefully and I am very familiar with related work.

---

> ### Author Rebuttal · Authors · 2023-08-29
>
> Thank you for your positive feedback and suggestions. We provide specific responses and clarifications as follows.
>
> > Reason 1: The ensemble improvement is not surprising.
>
> We’ll add more discussion on the difference between XoT and ensemble methods in the introduction section. While XoT builds upon various reasoning methods, our approach is similar to but different from the ensemble methods. In traditional ensemble methods such as majority voting (or self-consistency), each sample is executed multiple times following the same procedure. In contrast, XoT chooses different execution processes for different questions by prioritising the most possible method, which represents a more nuanced cognitive process akin to human reasoning.
>
> Additionally, we add the effectiveness and efficiency analysis between XoT and majority vote. Our findings indicate that XoT outperforms the majority vote approach, yielding a gain of 2.04 with a reduction of 16.7% in token count. A detailed description of the results is available in the response to [Reason 3 & Question 5] for Reviewer JpnC.
>
> > Reason 2: Insights on the planning module
>
> Thanks for your suggestion on the efficiency analysis, we will include it in the Analysis section! The design ethos of the planning module is for **efficiency** and **robustness**, instead of a huge performance boost.
>
> (1) For **efficiency analysis**, the planning module facilitates XoT to reach the final answer in fewer iterations by always starting from the most possible method. Specifically on GSM8K, XoT needs 1.46 iterations on average in comparison with 1.58 iterations with the fixed EPC order (EoT->PoT->CoT, the best performing fixed order). Specifically, 68.8% of the questions are resolved in the first iteration with XoT, as opposed to 57.2% when employing the fixed EPC order.
>
> (2) For robustness, although different fixed method orders demonstrate their respective advantages across various datasets, none of them excel across all datasets (Table 4). The planning module can dynamically adjust the execution order based on different questions, which ensures a more consistent and robust performance on average.
>
> > Reason 3: Analysis of active verification
>
> - _"Active verification seems to decrease the overall accuracy."_
>
> It appears that there might be a slight misunderstanding regarding the relationship between accuracy and active verification. We’d like to take this opportunity to clarify that the active verification leads to **overall performance gain**. We provide the analysis of the active verification in Line 453-462. Specifically, in Table 5, the ‘ACC’ indicates how well the verification module specifies the correctness of the answers, while the last column ‘XoT’ indicates the overall accuracy on the dataset. The inclusion of the active verification contributes to +2.3 gain on the overall accuracy. The main effect of the active verification is to decrease false positive rate (FPR), which enables XoT to try another method instead of keeping a false answer that passes passive verification (false positive samples).  We will revise the description for a clearer presentation of the results in Table 5.
>
> - _"consider showing the results without any passive or active verification in the ablation study"_
>
> It is a great suggestion to add more ablations regarding the verification module! Without the verification module, the framework cannot decide when to change to another method. Therefore, we choose the answer of the method suggested by the planning module for each question. We will add the detailed results of each dataset in the Ablation section. For a quick preview, on 9 datasets shown in Table 2, such ablation only achieves 79.33 on average, compared to XoT at 84.63. This indicates the importance of the verifications and method switching within the XoT framework.
>
> > Question 1: EoT implementation
>
> Sorry for the confusion on the implementation of EoT. Given the page limit, we present the implementation examples in Table 9 of the appendix on page 12. We will add more details about EoT in the main body of the paper. For hyperparameter details, we use 8 examples for each prompting and use greedy sampling in all runs.
>
> We appreciate your efforts in reviewing our paper and providing insightful suggestions! We hope our clarification above could provide a comprehensive understanding of our work and address your concerns.

---

### Official Review · Reviewer_JpnC · 2023-08-05

**Soundness:** 4

**Excitement:**

3: Ambivalent: It has merits (e.g., it reports state-of-the-art results, the idea is nice), but there are key weaknesses (e.g., it describes incremental work), and it can significantly benefit from another round of revision. However, I won't object to accepting it if my co-reviewers champion it.

**Missing References:**

Tree of Thoughts: Deliberate Problem Solving with Large Language Models. Shunyu Yao, Dian Yu, Jeffrey Zhao, Izhak Shafran, Thomas L. Griffiths, Yuan Cao, Karthik Narasimhan

The Art of SOCRATIC QUESTIONING: Zero-shot Multimodal Reasoning with Recursive Thinking and Self-Questioning. Jingyuan Qi, Zhiyang Xu, Ying Shen, Minqian Liu, Di Jin, Qifan Wang, Lifu Huang

**Paper Topic And Main Contributions:**

The authors propose XOT, a iterative framework that consists of three modules: planning, reasoning and verification to improve the LLM reasoning capability on mathematic problems. At each iteration, the planning model proposes a method, the reasoning model generates a solution based on the planning module and the verification verifies the solution. If the solution is incorrect the process is repeated again. XOT also assembles the three different prompting methods: COT, POT and EOT.

**Questions For The Authors:**

1. Can you provide some insights on the planning module. What

2. in the active verification, how often does the verification successfully predict the correctness of the answer.

3. can you also provide the performance of XoT with active verification in Table 3?

4. what happens if the verification module considers the answers from all 3 prompting methods as incorrect?

5. can you provide a distribution of the iterations to answer questions?

**Reasons To Accept:**

1. the paper is well written and easy to follow.
2. the motivation to combine different prompting techniques and perform verification is clear.
3. the experiments are thorough and clearly support the effectiveness of the proposed method.

**Reasons To Reject:**

1. On line 056- 061 the author mentioned they name an existing prompting method as EOT. On line 126-127, the authors claim to introduce EOT. However, the authors didn't clearly explain the details of EOT before the experiment section. My question is: (1) is EOT your new contribution? (2) if so, can you add details of the method? (3) If not, can you revise line 125 - 127?

2. The proposed verification method is limited and EOT is limited to some math problems. On table 3 all the datasets are verified with only passive verification and the authors didn't provide EOT performance.

3. Since this is a assemble method, stronger baselines should be considered. such as self-consistency chain of thought. What happens if you use take majority vote on three different prompting methods?

4. based on table 5 active verification can reduce the false positive rate and improve the performance but why you only use passive verification in table 3?

**Reproducibility:**

4: Could mostly reproduce the results, but there may be some variation because of sample variance or minor variations in their interpretation of the protocol or method.

**Reviewer Confidence:**

3: Pretty sure, but there's a chance I missed something. Although I have a good feel for this area in general, I did not carefully check the paper's details, e.g., the math, experimental design, or novelty.

---

> ### Author Rebuttal · Authors · 2023-08-29
>
> Thank you for your thoughtful and constructive feedback. We provide specific responses and clarifications as follows.
>
> > Reason 1: Clarification for EoT
>
> Sorry for the confusion on the naming of EoT. We name prompting methods that utilize equations to solve problems as EoT, in line with CoT and PoT. To address your questions:
> (1) Yes. EoT serves as a minor contribution. In our work, we employed distinct prompts in comparison to the concurrent work (He-Yueya et al., 2023).
> (2) Due to constraints on the page limit, we briefly introduced EoT in section 3.1 and showed its format in Figure 3. For a more direct illustration, we included a detailed example of EoT in Table 9 of the Appendix. We will add more details about EoT in the main body of the paper.
>
> > Reason 2: EoT on 'MATH' dataset
>
> We acknowledge the limitations of the EoT approach in addressing complex mathematical problems, e.g. on the MATH dataset. We did not include EoT as a method option, because most questions in MATH cannot be solved with an equation system, such as questions related to geometry:
> ```
> Using the side lengths 2,3,5,7, and 11, how many different triangles with exactly two equal sides can be formed?
> ```
> The above example inherently lacks the requirement for an unknown variable within an equation system to arrive at the answer.
>
> It is worth emphasizing, however, that such an exclusion does not in any way compromise the efficacy of the overall XoT framework. The evolving research community continues to produce novel prompting methods, which could potentially excel in the intricate subdomains of the MATH dataset and be further integrated in XoT. As designing new reasoning methods for specific datasets is out of the main scope of this work, we regard this exploration as a promising avenue for future endeavors.
>
> > Reason 3 & Question 5: Add majority vote as baseline & Distribution of the iterations needed
>
> For the convenience of response, we combine questions that share similar insights.
>
> Thanks for your suggestion! In response to your recommendation, we additionally conduct experiments involving the majority vote of three distinct methods. Our findings indicate that XoT outperforms the majority vote approach in a more efficient manner, yielding a gain of 2.04 with a reduction of 16.7% in token count.
>
> |               | Average ACC | #token / case |
> |:-------------:|:-----------:|:-------------:|
> | Majority vote |    82.59    |     5.4k      |
> |      XoT      |    84.63    |     4.5k      |
> |     Delta     |    +2.04    |    -16.7%     |
>
> Caption: #token / case denotes the average number of tokens consumed for one case (including prompts, question and response).
>
> We’ll add detailed results of each dataset in the experiment section. The vote is based on three answers generated by three methods (one answer per method). For a quick preview, on 9 datasets in Table 2, taking the majority vote of the three methods achieves 82.59 on average, while XoT achieves better performance at 84.63. Additionally, we observe that the majority vote fails on datasets containing questions that align exceptionally well with a specific method. Specifically, the majority vote achieves 79.73 on Algebra (most questions require equations to solve), while XoT achieves 89.94.
>
> The majority vote needs to execute all three methods to reach an answer, while XoT will stop when the answer passes the verification. We calculate the total token count as ``#total_token = #input_token + #output_token * 2``, according to OpenAI’s pricing policy. As shown from the table, XoT is able to achieve higher performance with a lower budget, exhibiting a reduction of 16.7% in expenses. The token count includes all the in-context examples used and is averaged across the number of the total questions in 9 datasets.
>
> - _“can you provide a distribution of the iterations to answer questions?”_
>
> Per your recommendation in Question 5, we find it would be very helpful to add a distribution of the iterations for further efficiency analysis. If we consider a round of planning-reasoning-verification as an iteration, XoT takes 1.46 iterations on average to reach an answer on GSM8K. In specific, 68.8% of the questions are answered in the first iteration, 15.8% in the second and 15.4% in the third iteration. We will integrate these analyses into section 5.2.
>
> > Reason 4 & Question 3: Active verification on 'MATH' dataset
>
> We will add more discussion on the active verification on MATH dataset in the revision. In general, we find adding active verification does not provide expected improvement on MATH dataset. The possible reason is that gpt-3.5-turbo model is not capable enough to apply assertion check on difficult questions in MATH. For example, it simply repeats the statements present in the provided code, which weakens the necessity of the active verification process. Since it can be mitigated by replacing a more capable backbone (e.g., gpt-4), we did not include the detailed discussion in the paper due to the page constraints. Moreover, 60.0% of the programs generated by PoT contain runtime errors - very few of them can proceed to active verification, limiting their feasibility for active verification.
>
> Per your request, we additionally add the active verification on the MATH dataset. Due to the time and budget limit, we conduct experiments on a random subset of 1000 out of 5000 examples on MATH. We try both gpt-3.5-turbo and gpt-4 for active verification on the PoT reasonings generated by gpt-3.5-turbo. With gpt-3.5 verifying, the false negative rate increases intensively (28.3) which negatively impacts the overall performance. Verifying with gpt-4 realizes a balance between the tradeoff of False Positive Rate (FPR) and False Negative Rate (FNR), leading to an increase in the overall XoT performance (outperforming CoT at 37.2, PoT at 26.5). With the availability of gpt-4, we will add more detailed results on each module in the experiments.
>
> |                    | Verification ACC  | FPR &#8595; |  FNR &#8595; | Overall XoT ACC |
> |:------------------:|:---------:|:------:|:-----:|:-----------------------:|
> | passive            |   83.8    | 20.68  |  3.77 |          43.4           |
> |   + gpt-3.5 active |   83.8    | 11.84  | 28.30 |          42.5           |
> |   + gpt-4 active   |   86.2    | 14.42  | 12.08 |          **44.4**           |
>
>  Caption: ‘Verification ACC’ denotes how well the verification module specifies the correctness of the answers. 'Overall XoT ACC' denotes the overall performance with the verifications involved in the whole framework.
>
> > Question 1: Insight on the planning module
>
> The design ethos of the planning module is for **robustness** and **efficiency**, instead of a huge performance boost.
>
> (1) For robustness, although different fixed method orders demonstrate their respective advantages across various datasets, none of them excel across all datasets (Table 4). The planning module can dynamically adjust the execution order based on different questions, which ensures a more consistent and robust performance on average
>
> (2) For efficiency, the strategic prioritization of the most promising method allows XoT to arrive at answers with fewer iterations. Specifically on GSM8K, XoT needs 1.46 iterations on average in comparison with 1.58 iterations with the fixed EPC(EoT->PoT->CoT) order. The efficiency is also indicated in the token count analysis above. We will add the clarification into the analysis section accordingly.
>
> > Question 2: how often does the verification successfully predict the correctness of the answer?
>
> In the last two rows of Table 5, we present the accuracy of each verification module. For PoT, the verification successfully predicts the correctness of 77.9% cases, while it achieves 81.2% for EoT answers. In addition, as indicated in Line 453 - 456, although marginally affecting verification accuracy, the inclusion of the active verification module plays a pivotal role in mitigating an initially alarming false positive rate, which contributes more to the overall performance.
>
> > Question 3: Please see the response of [Reason 4 & Question 3]
>
> > Question 4: What if all verifications fail?
>
> If the verifications fail on the first two methods, XoT will automatically choose the answer of the third method. That is, we do not apply verification on the last method.
>
> > Question 5: Please see the response of [Reason 3 & Question 5]
>
> We appreciate your kind reminder on the missing references, we will add them in the related work! We would be very grateful if you could reassess our paper based on our response and let us know if you might find our work more positive.

---

### Meta-Review · Area_Chair_QGgN · 2023-09-18

**Recommendation:** 5

**Metareview:**

This paper proposes XoT to improve math problem solving by LLMs by assembling a diverse group of reasoning paths with verification. Reviewers agree that while the proposed method is not exceptionally novel, it still presents a nice contribution to improving the reasoning capabilities of LLMs, and the experiments have demonstrated solid results. Based on the new results from the rebuttal, we strongly encourage authors to add experiments for more LLMs and reasoning tasks in their final version.

---

### Decision · Program_Chairs · 2023-10-07

**Decision:**

Accept-Main

**Comment:**

This paper proposes XoT to improve math problem solving by LLMs by assembling a diverse group of reasoning paths with verification. Reviewers agree that while the proposed method is not exceptionally novel, it still presents a nice contribution to improving the reasoning capabilities of LLMs, and the experiments have demonstrated solid results. Based on the new results from the rebuttal, we strongly encourage authors to add experiments for more LLMs and reasoning tasks in their final version.